# An Umbrella Review and Narrative Synthesis of the Effectiveness of Interventions Aimed at Decreasing Food Prices to Increase Food Quality

**DOI:** 10.3390/ijerph16132346

**Published:** 2019-07-02

**Authors:** Chiara Milani, Chiara Lorini, Alberto Baldasseroni, Claudia Dellisanti, Guglielmo Bonaccorsi

**Affiliations:** 1School of Specialization in Hygiene and Preventive Medicine, University of Florence, 50134 Florence, Italy; 2Department of Health Sciences, University of Florence, 50134 Florence, Italy; 3Department of Prevention, Local Health Unit Toscana Centro, 50135 Florence, Italy

**Keywords:** umbrella review, diet sustainability, food price, fiscal intervention

## Abstract

Background: sustainability of population diet is a public health concern: the high price of healthy food is one of the main causes of diet-related health problems. The aim of this study is to synthesize the evidence produced by systematic reviews that evaluated the effectiveness of decreasing healthy food prices to improve accessibility in order to positively modify the dietary pattern. Methods: We carried out a review of systematic reviews that examined the effects of the interventions, by exploring the online databases PubMed, Embase, Web of Science, Cochrane Library and hand-searching the reference lists. Results: after screening by titles and abstracts, we selected 11 systematic reviews that met the inclusion criteria, plus one that was hand-searched. The review generally presented a good quality. Studies concluded that measures aimed at modifying the prices of targeted healthy food were effective in improving population diet by modifying what people buy. Conclusions: the complexity of the outcome—population diet—as well as the poor transferability of data across populations and geographical areas makes it obligatory to provide clear and universal conclusions. Nonetheless, this should not stop policymakers from adapting them and resorting to food fiscal interventions to improve people’s diet and health.

## 1. Introduction

Sustainability of the food system, which depends on its environmental and economic impact, as well as on its role in individual and community health, represents a fundamental issue in the public health nutrition area, and is an important issue in high (HICs) as well as in low-–middle income countries (LMICs).

Food production is responsible for 20–30% of greenhouse gas emissions, soil consumption and erosion, 70% of total water consumption, loss of biodiversity, deforestation, air and water pollution [1]. Moreover, in association with the demographic transition and increase in life expectancy, we are facing a global nutritional transition towards a so-called occidental diet, rich in processed foods, meats, oils, refined sugar, and fats [2]. This dietary shift in the global population—in the first as well as in the LMICs—is responsible for an increase in the incidence of chronic non-communicable diseases [3].

Socio-economic status influences diet inequalities and, consequently, health inequalities between HIC and LMIC, and also within HICs because of the different prices of nutritious, low energy-dense foods, such as fruits and vegetables, and poor-nutrient, energy-dense foods [4]. Consequently, food (un)sustainability can give rise to food insecurity. It is generally based on three pillars: lack of availability, lack of access, and lack of utilization, which are hierarchically linked to each other [5,6]. The insecurity related to insufficient access to healthy and sustainable diet continues to represent a huge problem in the HIC, too, because of the high price of high-quality food. Since energy-dense food high in saturated fat, trans fats, sugar, and salt is cheaper than the healthier ones, absolute or relative poverty, low income and/or price volatility negatively affect the dietary pattern adopted by the most disadvantaged part of society, despite the availability of healthy foods [7]. In fact, economic access is strongly related to the price of food and to family income. Economic access to food depends both on food cost and price, and on the percentage of family income devoted to food purchase [6]. Food prices influence what people buy [8]: in this perspective, fiscal policies aimed at lowering prices to increase affordability for all, could influence consumers’ purchases and diet modification, promoting healthy and sustainable food consumption. The interventions that act on prices and affordability are classified among those that change the market environment, together with meal regulations at school or at the workplace and could be addressed to the general population or only to disadvantaged consumers [9].

The World Health Organization (WHO) stated that fiscal policies that act on healthy food accessibility through incentives on purchases, especially subsidies for fruits and vegetables, could be effective in shifting dietary pattern with positive health outcomes [10].

Between 2006 and the first half of 2018, many systematic reviews (SRs) assessed the effectiveness of a decrease in the price of healthy food to enhance accessibility for different income subgroups of the population. However, to the best of our knowledge, we have not come across any systematic review of systematic reviews that highlights the limits of results and evidences coming from the studies included.

Hyseni [11] made a non-systematic review of reviews of public policies to improve population diet in various contexts to identify the policies that work better. The author concluded that specific taxes and subsidies induce respectively a decrease in the consumption of sugar-sweetened drinks and an increase in the intake of fruits and vegetables. Fiscal policies at supermarket seem to be more effective than nutrition education interventions. However, the studies included in the review do not report data and results either on the relative effectiveness of every single specific type of measure, or on how to implement them.

The aim of this study is to seek and address the evidence provided by previous SRs in order to get a fair idea of the state of the art to show that decreasing the price of healthy food increases access and, therefore, has a positive effect on modifying dietary patterns. Moreover, we also aim to stress the limits of the data produced through study designs that perhaps are not appropriate for exploring complex phenomena such as diet and dietary patterns.

## 2. Materials and Methods

The study has been done following Aromataris et al. in Joanna Briggs Institute Manual to realize an umbrella review [12]. It is a systematic review of systematic reviews. The methodology used is different from that of Cochrane [13] for an overview of reviews, because we included not only Cochrane intervention reviews produced by individual Cochrane review groups, but also other reviews that met the inclusion criteria, and is, therefore, more inclusive.

### 2.1. Inclusion Criteria

We used the PICOT scheme (population, intervention, comparison, outcome, and type of study) (Table 1) to identify the papers meeting the inclusion criteria: (1) peer-reviewed studies; (2) studies on healthy adults; (3) studies assessing the effectiveness of fiscal policies; (4) studies measuring the change in the consumption of healthy food or in the purchases at the point of sale; (5) type of studies: SRs.

We included the SRs that had examined the effects of fiscal interventions aimed at decreasing the price of healthy (low-fat or low saturated fats) foods, such as subsidies, incentives, coupons and vouchers, food discounts, cash rebates on purchases, and any other form of advantage. Studies also aimed at analyzing the effectiveness of other fiscal interventions, such as the hyper-taxation of unhealthy foods, were included in the final review only where it was possible to assess separately the effects from incentives and those from taxes. 

We assessed only studies on adult healthy people. We did not include studies on vending machines. These criteria were assumed to assess the effect of interventions on active, healthy consumers, whose purchasing choices were modifiable by cost saving actions that made it easier to acquire healthy foods. 

We did not include studies not written in English, not published in peer-reviewed journals, and book chapters, theses, and abstracts. Moreover, we decided not to insert a data range in the search process. 

### 2.2. Search Strategy 

We searched articles in the following online databases: PubMed, Embase, Web of Science, and Cochrane Library. We also hand-searched the reference lists without any time limits.

To build the search query, we included in the final strings each of the following terms about interventions: ‘fiscal policy’, ‘public policy’, ‘tax’, ‘subsidy’, ‘incentive’, ‘price’, ‘voucher’ and combined them with the following terms that refer to the study outcome: ‘healthy diet’, ‘sustainable diet’, ‘healthy food pattern’, ‘food consumption’.

In the PubMed database, we added the term ‘Meta-Analysis [ptyp] OR systematic [sb]’, to select only SRs. Table 2 shows the complete search strategy used for PubMed database. The PubMed search strategy was then adapted for the other databases: it is available from the authors upon request.

### 2.3. Study Selection and Data Extraction

The search query identified 688 articles (496 in PubMed, 153 in Web of Science, 38 in Embase, 1 in the Cochrane Library). After screening all the articles for titles and abstracts, 34 articles were selected, and 19 after removing duplicates. Eight were excluded since they did not meet the PICOT inclusion criteria (Figure 1). In the end, 11 SRs met the inclusion criteria plus one that was found through hand search. Figure 1 represents the PRISMA (Preferred Reporting Items for Systematic Reviews and Meta-Analyses) flow-chart process [14] of study selection. Two researchers reviewed the papers independently.

We also assessed the quality of the included reviews, using the score assigned according to the Health Evidence tool, both when it had already assigned and utilizing the same criteria for rating the others by means of the same criteria [15]. Each study received a score in the range from zero to 10: a score of four or less meant a weak study quality; a medium quality if the score was 5–7; high quality if it was 8–10. The score quantified the strength of the data in the studies included in each review and was not an inclusion criterion.

## 3. Search Results

### 3.1. Description of the Selected Studies

The total number of primary studies included in the 12 reviews was 344. For each review, the studies that met the PICOT criteria varied from one to 19, for a total of 111. 

Table 3 presents the characteristics of the included reviews: year of publication, number of studies included in the review, and number of studies, in each review, that meets the PICOT (study design, sample size, population age, setting, specific type of intervention, with the inclusion criteria described before), number of databases sourced and searched, date range of database search, participant details, the setting and context to standardize the data extraction process for each study included.

Table 4 explains the conclusions of each review, together with the comments, notes and limits observed by the umbrella review authors regarding any included study and review as such.

The types of studies included in the SRs were 37 modelling studies [16,17,18,19] and 74 field studies—54 experimental [18,19,20,21,22,23,24,25] and 20 empirical studies [21,22,26,27]. In all the reviews there are some articles that don’t meet the PICOT criteria, all the included studies met the PICOT criteria.

### 3.2. Findings

Interventions examined in the review could be classified as individual incentives (redeemable coupons, vouchers) and population incentives (discounts, cash back rebates, and reduction in value-added tax (VAT)) to a range of healthy food or to a specific, single food set at point of sale (supermarkets, farmers’ markets, organic food shops) and in restaurants and cafeterias. In many SRs, most studies were about low-income populations, specifically women. Given this huge heterogeneity, both in the studies included in each review and among the different reviews, it was not possible to classify the studies according to a single sub-type of intervention, and to establish a strong score through the comparison of the agreement of the results in the reviews. Moreover, the definition of very specific criteria for the PICOT inevitably allowed the inclusion of only a few studies from each selected SR. For this reason, we had to create a ‘narrative overview of the results’ of the studies included. The reviews, while concluding almost unanimously on the effectiveness of incentives in changing food habits towards a healthier diet, could not establish the strength of the different types of interventions separately.

Based on the results coming from the different study designs included in the SRs, it was not possible to argue the level of evidence of the different measures emerged. Results are presented in relation to some variables of the primary studies included in the SRs, in according with the SRs results. These variables are: the presence of an economic threshold in the fiscal intervention and the presence of a follow-up period.

Concerning the threshold, studies do not agree in affirming the presence of a threshold of fiscal advantage from which an intervention is effective in modifying the outcome (purchase or consumption). A lower threshold represents the minimum level to observe effectiveness. They all observed a one-way relationship between the level of price decrease and the amount of consumption increase: the greater the economic advantage, the higher the positive effect. Moreover, the result is better where a combination of subsidies and other interventions are used. 

In modelling studies [16,17,18,19] the threshold established for having a considerable effect was from 10 to 20% and the effects varied from 2.5 to 25%. Only in one SR [17], the level was smaller: for 1% decrease in fruits and vegetable prices, the increase in the purchases was 0.35%. Considering the experimental studies [18,19,20,21,22,23,24,25], the results do not describe the achievement of any important goal: only one SR [24], based on experimental studies, had a level of evidence A, class I, using the AHA (American Heart Association) evidence grade system, and stressed a change in consumption from 12.5% to 16% for a 10% price decrease when subsidies were given to improve diet.

In one study [16], the researchers affirm there is evidence of taxes and subsides ensuring a price decrease of more than 10–20% on food price, and the higher the economic advantage, the higher the purchase of healthy foodstuffs. A similar conclusion has been made by other research [24], which states that a 10% decrease in healthy food prices increases its consumption by 12% (95% CI = 10–15%; *N* = 22 studies). Regarding the main food groups, the subsidies increase fruit and vegetable consumption by 14% (95% CI = 11–17%; *N* = 9), and by 16% in case of other food groups (95% CI = 10–23%; *N* = 10). Eyles [17] affirms that a 10% decrease in fruit and vegetable prices could increase their consumption by 2.1–7.7%. In another study [21], the conclusion was that a price decrease of 10% to 20%, both in modelling studies and in trials, is effective. In [22], the threshold of subsidies varies between 10% and 50%, and $7.50 and $50 for vouchers. Another study [18] concluded that the threshold that ensures consistent success is 10–20%, even though the review reports studies in which subsidies range from 1.8% to 50% and all found a success (from 2–5% to 25%). The bigger the amount of the subsidy (or of the combination of interventions), the more positive was the effect. Another review [19] identified a minimum level (threshold) of 10–15% beyond which taxes and subsidies can be effective. 

Only in four reviews, the primary studies had a follow-up period of assessment [21,23,26]. A study [22] highlights the conflicting results from studies with a follow-up. Also in Purnell [23], where a follow-up is present, the evidence of success is not maintained. In Black [21] the positive effect is detectable only in two studies after the follow-up period. A randomized controlled trial shows an increase in purchases of fruits and vegetables after six months with a 12.5% decrease in the food price, and the effect, though attenuated, persists even at 12 months. The intervention on other healthy foods indicates similar results. Another one [26], a controlled before-after study, concluded that the effects of subsidies were maintained after a six-month follow-up period.

To summarize the different conclusions of the SRs, no study states that a threshold lower than 10% of the food price discount can have any success. Nevertheless, many studies used a fixed, non-variable, level of subsidy that does not let us identify a dose-response relationship for different populations and time spans. Five reviews [20,21,23,25,26] reported the absence of enough information to establish an effective threshold and a dose-response relationship.

Moreover, a paradoxical effect can emerge from the price reduction of healthy foods, that is a “diversion” of money towards fats, calories and unhealthy foods [17,18].

### 3.3. Quality Assessment of the Studies

About the quality of the studies (Table 4), the use of the above-mentioned criteria showed that no review had a quality score of less than 5 (low quality), testifying to a comprehensively good quality of the chosen reviews.

## 4. Discussion

The attempt to implement a systematic review of systematic reviews was prompted by a need to summarize the effectiveness of interventions of food price decrease in modifying population behaviour regarding the purchase and consumption of healthy foods and dietary patterns. The aim was to provide indications for policymakers to carry out policies that improve access to healthy foods and promote appropriate food choices. To fulfil this aim and to make a better use of and improve data, we decided to resume suggestions and recommendations and then to highlight the limits of the collected results in a complex field as population nutrition and dietary choices, since they sometimes pose an obstacle to the implementation of such interventions to make them structural, long-acting measures. 

One SR [21] stated the practical and ethical limits of trials, which are intrinsic to all public health interventions with the need of long-term evaluation and economic analyses. In another [17], the authors affirmed that the best evidence for the success of the interventions of food price might come from ‘natural experiments’. One review [22], although concluding that subsidies reducing healthy food prices succeed in improving population diet reducing healthy food prices, showed awareness of the small and convenience samples from which results come, the short duration of the interventions and follow-ups, the lack of cost-effectiveness studies, as well as impact measures on food industry, and finally the lack of overall diet assessment. What emerges is a clear need to design studies able to explore these aspects at population level. A paper [19] suggests that to do this, it might be necessary to prepare population-wise fiscal policies to evaluate their outcomes and effects after an adequate period of experimentation before actually implementing them. Based on these statements, we analyzed the limits of the studies.

### 4.1. Limits of the Studies Included in the Review

We analyzed the SRs that included studies that differed in their study design, targeted population (for characteristics and sample size), setting of implementation schedule, duration, and a follow-up period. These differences in the primary studies included in the systematic reviews represented serious obstacles in realizing the aim of this umbrella as described in the following paragraphs.

First, the lack of a clear definition of criteria to make such heterogeneous studies comparable: it is sometimes impossible to infer the ‘real’ impact of each experience and compare the contribution obtained in each of the selected researches. Furthermore, though the primary studies usually show a success in the expected outcome (except for three studies [25]), the authors did not report any follow-up assessment in the long term.

Regarding the study designs, reviews include field studies (randomized controlled trails and other experimental studies) and modelling studies. Even if randomized controlled trials are methodologically robust, they cover a short period of time, select a small sample size, and their external validity is a matter of debate. On the other hand, modelling studies indicate the potential effect of an action by utilizing data coming from the entire population, but they are less effective in showing the real effect of the measure. Therefore, in both study designs, there is a real difficulty in measuring the strength of the intervention in the long term, which would be the major outcome of an appropriate dietary behaviour. Change habits—like food choices—needs a long time to be observed, measured and to affirm that the interventions succeeded. Once this is established, it is possible to somewhat increase prices or stop with an information campaign. Moreover, a short duration does not allow researchers to observe the expected and unexpected unintended consequences of the interventions, such as the decrease in sales of healthy food with an unchanged price after the lowering price of another healthy food. 

Regarding an increase in healthy food consumption, most actions examine the impact on specific targeted foodstuffs and not on the total calorie intake and dietary pattern. Some studies measure the outcome using both the sales records and the change in self-reported consumption. Even if the first is a proxy of the second, they are not the same thing [22]. 

Moreover, few studies consider the assessment of interventions by adjusting for socio-economic status. 

Finally, we found very few instances of cost-effectiveness [19] and the equity impact of the interventions. 

### 4.2. Limits of Our Review 

An umbrella review itself has its limitations in the methodological process, such as the potential loss of information because of an excess of the synthesis of already produced reviews. 

The choice of selecting only systematic reviews that include interventions aimed at buying and consuming healthy foods (especially fruits and vegetables) prevented us from assessing the complex effect arising from a combination of different interventions and any other unexpected effect. Moreover, interventions aimed at producing effects on people’s health can be assessed only over very long periods. 

Another possible source of bias of this umbrella review is the use of a limited number of databases, which can lead to a potential exclusion of some relevant studies. 

Thus, the limitations can be attributed to the quality of the included primary studies on which systematic reviews are built, as stated by the authors, as well as the strength of the conclusions of systematic reviews themselves, and our umbrella review, too. Even if the search query contained the term ‘sustainable’ as an outcome measure, the word produced no results in the systematic review search.

The choice of the outcome measure—an increase in the purchase or consumption of healthy foodstuffs, and no other health outcomes—was dictated by the need to detect the actions that could be more effective in building a healthier as well as a more sustainable diet nationwide. This ‘imposed’ limit is justified by the fact that, on the one hand, there is evidence of and consensus about the impact of food production on the environment [1] and, consequently, the need to modify people’s diet in a more sustainable way [28], and, on the other, actions implemented to produce a dietary change tend to be assessed and judged ‘only’ through a change in the purchase and/or in the consumption of healthy food, but not in relation to the sustainability of the food production, delivery, and sale. Although there is a current debate in the literature on the definition of sustainable diet—and many authors identify the Mediterranean diet as an example [29] of it—sustainability is not often a considered and promoted dimension in the implementation and impact assessment of the interventions aimed at modifying diet.

A large number of scientific studies have become available in the last ten years on the effectiveness of actions aimed at modifying food consumption in different settings (supermarket, small shops, farmers’ market, restaurants, fast food, and café) and in the healthy adult population.

We observed that systematic reviews included different types of primary studies, such as field (i.e., randomized controlled trails, cross-sectional, quasi-experimental) as well as modelling studies. We tried to discuss the results, strengths, and weaknesses of both types, when highlighted by the same authors or derived by the study design itself. The need to give relevance to these weaknesses raised by the attempt of underlining the difficulty in designing a study that deals with a complex matter as diet, at a population level. Field studies produce robust results from a methodological point of view, but they are usually too short to indicate long-term effects on dietary patterns as well as health outcomes, and sample size is often too small for the results to be applicable nationwide. On the other hand, modelling studies let us widen the sample size and the number of variables to be included that could determine the success of the interventions, but they are, nevertheless, only a representation of the real world, not ‘the real world’ and the ‘real life’ of the consumers. Only three systematic reviews included nature-oriented studies [19,21,26]: in this perspective, natural experiments are an ad hoc study design adopted when researchers have a strong expectation of success in a proposed intervention, but lack supporting scientific evidence and when randomized controlled trials are not feasible [30]. Among these, interrupted time-series studies—though considered only in one review—are important because they can support the causal inferences from natural studies. They were, in fact, ‘analysis where data had been collected at three or more time points both before and after an intervention was implemented’ [21]. 

As a matter of fact, we cannot expect trials to be completely exhaustive if they are applied to produce evidence based on public health and public policies in the social context, because they tend to consider and measure only specific and targeted outcomes more than analyzing the many different features of a public health problem [31]. Randomized controlled trials and systematic reviews, mostly based on randomized controlled trials (one [20] included only randomized controlled trials, in others [24,25] the majority of studies were experimental studies), can hardly be assessed about their external validity [32] even when they show the success of interventions by means of a robust statistical evidence and without selection bias. 

Moreover, taking decisions on public health policies needs results and supports other than clinical outcomes only, for example, dimensions such as economics, sociology, human rights, equity, anthropology, to take decision that consider the needs of all the people independently from cultural differences and acceptability [33]. Randomized controlled trials are not the best study design to consider these fields of knowledge, because they often insert an intrinsic limit to the results: they tend to hide those social structures, which at the same time influence the success of a specific policy and are the agents of those mechanisms that the intervention wants to remove. Let us consider, for instance, disparities among groups based on income, education, social organization, and, in general, all the factors linked to people’s living environment. Even though their results show effectiveness in modifying people’s diet, ignoring the effect on socio–economic inequalities lead to conclusions that hardly permit the creation of a decision-making political process. Consequently, they cannot be ignored in an attempt to produce evidence in this field. So, the lack of evidence or of sufficient elements to design policies does not refer to the weakness in the design of experimental studies, but the need to include these other fields of knowledge to answer doubts, questions, and fulfil the need for information in implementing any ‘structural’, long-term actions. It is possible to implement and assess the intervention of decreasing healthy food prices population-wide, together with other—not market-based—supporting actions. Evidence from the studies shows reasonable elements of success.

In general, studies conclude that measures modifying the prices of targeted healthy foods are effective in terms of greater affordability and, as a consequence, determine a shift in decisions regarding what to buy [34,35]. 

## 5. Conclusions

In conclusion, this is the first ‘umbrella review’ to make a synthesis of the systematic reviews produced on the effectiveness of interventions of price reduction to provide a change in population dietary pattern in a more sustainable, as well as a healthy, way. Fiscal policies that address accessibility through incentives on purchases, especially subsidies that decrease fruit and vegetable prices, and taxes on sugar-sweetened beverages, could be effective in modifying consumers’ choices towards a more appropriate dietary behavior. However, our work has not been able to identify population-wide fiscal interventions on food prices than can be successfully implemented in different settings. In fact, the presence in the reviews of studies with different interventions and outcomes meant the impossibility of applying—as well as measuring and comparing—the same intervention in different conditions, populations, and contexts. In this umbrella review, this limitation emerged strongly.

Public food policies represent specific actions whose implementation is not separable from the social context. The lack of evidence to be provided to policy-makers to put them in practice does not depend only on the weakness of studies, but is also linked to the complexity of the phenomenon—population diet—as well as to the poor transferability of evidence produced for other aims and field of knowledge.

In spite of the mentioned limits, it emerges that fiscal policies are effective in increasing access to healthy foods, so promoting appropriate food choices. This represents a clear message to policymakers to act by means of structural, long-acting fiscal measures together with educational interventions aimed at preparing population to adopt different food choices.

The effect of the implementation of these policies should be assessed by high quality study designs with very long lasting follow up periods, and considering many different effects, both positive and negative, regarding food choices and behaviors of the populations. 

## Figures and Tables

**Figure 1 ijerph-16-02346-f001:**
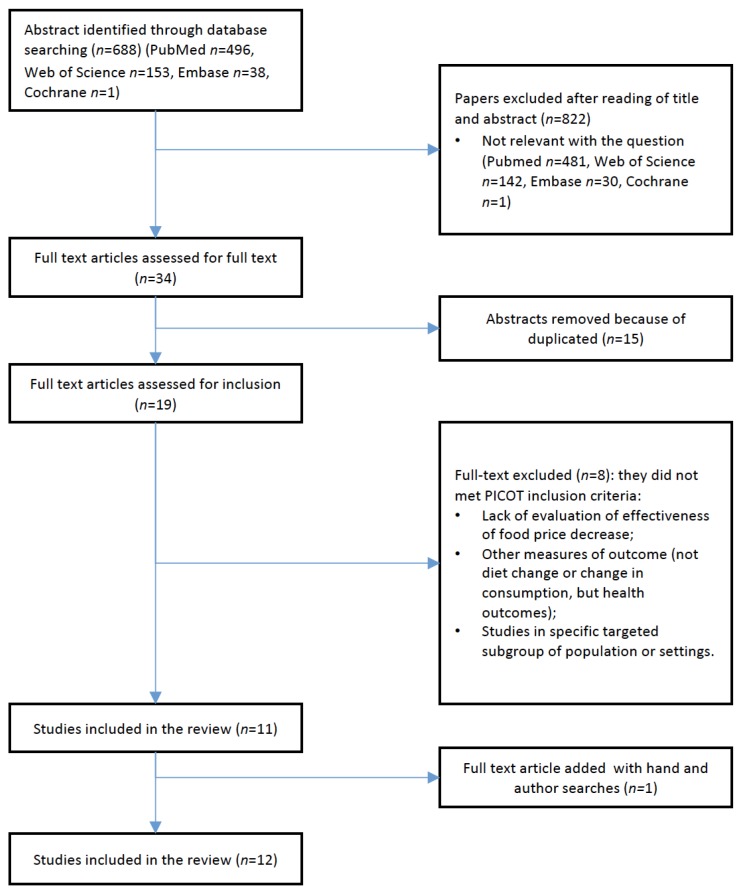
PRISMA (Preferred Reporting Items for Systematic Reviews and Meta-Analyses) flow diagram.

**Table 1 ijerph-16-02346-t001:** Population, intervention, comparison, outcome, and type of study (PICOT) scheme to define inclusion criteria.

Parameter	Description
Population	Inclusion: healthy adult, general population. Exclusion: children, adolescents, unhealthy people.
Intervention	Inclusion: price decrease at the point of purchase (discounts, coupons, cash rebates) or other forms of incentives on healthy foods (fruit and vegetables and low-fat foods) purchasing and at restaurant, cafeterias Exclusion: other fiscal policies (taxes on energy-dense food high in saturated fat, trans fats, sugar and salt).
Comparison	Lack of intervention
Outcome	Inclusion: increase of purchase at point of sale, or of consumption of healthy food. Exclusion: health outcomes (weight loss, chronic disease…).
Study design	Inclusion: systematic reviews.

**Table 2 ijerph-16-02346-t002:** Description of the search strategy in PubMed.

Search ID	Terms	*N*
1	Fiscal polic *	173
2	Tax	13,643
3	Public policy * OR tax (1 OR 2)	54,441
4	Subsid *	22,371
5	Fiscal polic* OR tax OR subsid * (3 OR 4)	35,752
6	Incentive	204,512
7	Fiscal polic * OR tax OR subsid * OR incentive (5 OR 6)	239,510
8	Price OR pricing	327,865
9	Fiscal polic * OR tax OR subsid * OR incentive OR price OR pricing (7 OR 8)	556,367
10	Voucher	1069
11	Fiscal polic * OR tax OR subsid* OR incentive OR price OR pricing OR voucher (9 OR 10)	557,139
12	“healthy diet”	4394
13	(fiscal polic * OR tax OR subsid * OR incentive OR price OR pricing OR voucher) AND (“healthy diet”)	461
14	Healthy diet	31,806
15	(fiscal polic * OR tax OR subsid * OR incentive OR price OR pricing OR voucher) AND (“healthy diet” OR healthy diet)	1694
16	"sustainable diet"	61
17	(fiscal polic * OR tax OR subsid * OR incentive OR price OR pricing OR voucher) AND (“healthy diet” OR healthy diet OR “sustainable diet”)	29,452
18	Sustainable diet	1150
19	(fiscal polic * OR tax OR subsid * OR incentive OR price OR pricing OR voucher) AND (“healthy diet” OR healthy diet OR “sustainable diet” OR sustainable diet)	1698
20	Healthy food pattern	2430
21	(fiscal polic * OR tax OR subsid * OR incentive OR price OR pricing OR voucher) AND (“healthy diet” OR healthy diet OR “sustainable diet” OR sustainable diet OR healthy food pattern)	1818
22	Food consumption	101,450
23	(fiscal polic * OR tax OR subsid * OR incentive OR price OR pricing OR voucher) AND (“healthy diet” OR healthy diet OR “sustainable diet” OR sustainable diet OR healthy food pattern OR food consumption)	14,375
24	(Meta-Analysis[ptyp] OR systematic[sb])	366,291
25	(fiscal polic * OR tax OR subsid * OR incentive OR price OR pricing OR voucher) AND (“healthy diet” OR healthy diet OR “sustainable diet” OR sustainable diet OR healthy food pattern OR food consumption) AND (Meta-Analysis[ptyp] OR systematic[sb])	496

* it is commonly used as a wildcard symbol to broaden a search by finding words that start with the same letters, so as to retrieve variations of a term with less typing.

**Table 3 ijerph-16-02346-t003:** Description of included review characteristics (RCT: randomized controlled trial; VAT: value added tax; CBA: controlled before-After; WIC: women, infants, and children; ITS: interrupted time series; SR: systematic review; BMI: body mass index; DM2: diabetes mellitus type 2).

Author, Year	Number of Studies (Tot and Subsidies Only)	Databases	Population—Setting—Intervention
Thow, 2010 [16]	24 studies (18 modelling studies, six empirical–ecological studies), four included (all are modelling studies)	Medline, ProQuest and Business Source Premier academic databases and Google Scholar.	Defiscalization (decrease of VAT) of fruit and veggy foods.
Eyles, 2012 [17]	32 studies, 15 included (all are modelling studies; seven about subsidies, eight combinations of tax and subsidy)	Medline, Embase, and Food Science and Technology Abstracts between 1 January 1990 and 24 October 2011.	Subsidies on targeted foods (fruits and vegetables (F&V), soft drinks, F&V and fish, fibre) and on a range of healthier products and combination of tax and subsidy (total fat and/or saturated fat tax and a fibre/grain and/or fruit and vegetable subsidy).
Thow, 2014 [18]	38 studies, 14 included: one RCT and 13 modelling studies	MEDLINE, Web of Knowledge, EconoLit, Business Source Premier academic databases and Google Scholar (the first 15 pages of each search using Google Scholar), between January 2009 and March 2012.	Subsidies (most in supermarket and in combination with taxes) on targeted foods or on a wide range of healthy foods.
Niebylski, 2015 [19]	78 studies, 16 included: five modelling studies (two SR of modelling studies), three observational studies, (two SR), eight experimental studies.	PubMed, Medline and Cochrane Library databases (between June 2003 and November 2013) and Google Scholar (between June and November 2013)	Subsidies (most about discounts) in supermarkets on targeted food or “healthy food”, evaluating healthy food purchases (fruit and vegetables) and increased consumption.
Wall, 2006 [20]	Four RCT studies (about incentives), one included (three excluded because of setting—work or school setting, outcome –weight loss, population—obese adult)	MEDLINE (1966 to April 2005), EMBASE (1980 to 2005), CINAHL (1982 to April 2005), Cochrane Controlled Trials Register/Library (to 2005), and PsycINFO (1972 to April 2005) databases.	Coupon to purchase food at the farmer’s markets given to low-income women belonging to WIC programme.
Black, 2012 [21]	14 studies, seven included: three RCTs, three CBAs and one ITS. (seven with different outcome: weight at birth, mother’s biomarkers and BMI…).	Medline, Cochrane, DARE, Embase, Cambridge Scientific Abstracts—Social Services Abstracts and Sociological Abstracts, Web of Science- Science Citation Index, Social Science Citation index, CINAHL, Informit-Health, Food Science and Technology Abstracts, and EconLit. Between 1980 and November 2010.	Subsidies alone or in combination with other intervention to socio-economic disadvantaged families, most in the WIC program (vouchers or discounts to a wide range of healthy food or only to fruit and vegetables or juice); the other about supermarket price discounts.
An, 2013 [22]	20 studies, 15 included (seven7 RCTs, five CBAs, three cohort). (one in South Africa).	Cochrane Library, EconLit, MEDLINE, PsycINFO and Web of Science	Subsidies (price discounts or vouchers) in Supermarkets and farmers’ markets (9), cafeterias (5) and restaurants (1), to adolescents or adults (metropolitan transit workers and low-income women), measuring both sales and self-report intake.
Purnell, 2014 [23]	12 studies, two included (two quasi-experimental) (10 excluded because outcome is weight loss)		Vouchers to WIC low income women
Afshin, 2017 [24]	30 studies, 13 included: experimental studies (four RCTs, nine non-randomized) (seven excluded because about vending machines or school and work setting and BMI as outcome)	PubMed, Econlit, Embase, Ovid, Cochrane Library, Web of Science, and CINAHL.	Cash back rebates, coupons and discounts at point of sale (supermarkets/markets), in communities and in restaurants/cafeterias.
Gittelsohn, 2017 [25]	30 studies, 19 included: experimental studies (13 RCTs, six quasi-experimental). (11 excluded because about sub-group of population: workers, obese women with DM2, children at school or about taxes on unhealthy foods).	MEDLINE, Embase, PsycINFO, Web of Science, ClinicalTrials.gov, and the Cochrane Library—from January 2000 through December 2016 (6 electronic databases).	Discounts, coupons/vouchers most to people of Food assistance programme and cash rebates at the point of sale and at all retailers in a
Alagiyawanna, 2015 [26]	18 studies, only two included: 1 natural experiment and 1CBA	Medline (OvidSP) (1946-present), PubMed, EconLit and PAIS (Proquest), Global Health (OvidSP) [1973-present], Global Health Library.	Additional Vouchers to Post-partum low-income women of the WIC programme; reduction in soft drinks tax.
Hillier-Brown, 2017 [27]	30 studies, three included: one cohort study, two CBAs (27 excluded because about other interventions)	ASSIA (ProQuest), CINAHL (EBSCOhost), Embase (Ovid), MEDLINE (Ovid), NHS EED (Wiley Cochrane), PsycINFO (EBSCOhost), from January 1993 to October 2015	Redeemable coupons (alone and associated with health promotion in restaurants and fast food) in food outlets that sold “ready-to-eat” meals and are openly accessible to the general population to increase intake or purchases of healthy food.

**Table 4 ijerph-16-02346-t004:** Description of included review characteristics (RCT: randomized controlled trial; SES: socioeconomic status; AHA: American Heart Association; F&V: fruits and vegetables).

Author	Conclusions	Limits	Health Evidence Score
Thow, 2010 [16]	Taxes and subsidies can influence consumption, particularly when they are large (at least 15% of product price).	High number of modelling studies, many of them about effect on targeted foods, not the overall diet. No experimental studies available. Empirical studies included had limited sensitivity. Narrative summaries. Conclusions not sufficient to establish a threshold of effectiveness, resulting in not sufficient information to implement a public policy.	6
Eyles, 2012 [17]	Price interventions are able to modify consumption. Price interventions have success in modifying consumption. There is a linear relationship between the value of the subsidy and the increase of fruit and vegetables consumption, i.e., 1% of decrease in the F&V price increase purchase of the same food-group of 0,35%. Eight modelling studies evaluate the association of taxes and subsidies: major evidence of success with a risk of compensatory purchasing.	All studies are modelling studies. Low quality of studies included (27/32): heterogeneity in model used and value of the subsidy. Lack of evidence in low-income countries. Only four studies estimated a health benefit for lower socio–economic population compared to high, even if the majority of these studies (11/14) estimated that fiscal policies would result in absolute improvements in dietary outcomes for low-income.	9 *
Thow, 2014 [18]	Strong evidence from robust modelling studies and from one RCT of taxes and subsidies combined, even if there is evidence that subsidies can increase caloric intake. Threshold from which success is observed is 10–20% (in one study a price decrease of only 1.8% resulted effective) of the food price. The bigger the amount of the subsidy (or of the combination of interventions), the bigger the positive effect (from 2–5% to 25%).	Inclusion of different type of studies allows assessing strength and limits in producing results of all the different studies. Heterogeneity of population group studied made the conclusion uncertain.	6
Niebilski, 2015 [19]	Consistent moderately strong evidence of success, concluding the support to the implementation of interventions population-wide. Modelling studies: maximum success in association of taxes and subsidies of a minimum of 10–15% of the food price. Association with fiscal policies with other actions (food education or marketing actions) enhance the success.	Many results based on modelling studies and price elasticity rather than field works. Most experimental studies are localized and not demographically representative. No engaging of food industry in their will of implementing the actions with the need to engage many stakeholders in the design of an action within the entire population.	7
Wall, 2006 [20]	Positive effect on both food purchase and weight loss: incentives that decrease price at the point of sale are more effective in modify consumers’ choices than individual interventions (such as coupons).	Results are not sufficient to assess how to implement the interventions population-wide. Only four RCTs included. Short duration (max 18 months). Follow-up only in one study. Lack of cost-effectiveness measures and impact on food industry. Lack of evaluating the effect in different subgroup of population (ethnicity, SES…).	8
Black, 2012 [21]	Success of intervention but in limited high-quality studies. Modelling studies: female participants of the WIC program in USA demonstrated a 10–20% increase in targeted nutrients and foods due to the subsidy program. Studies about other actions showed similar improvements in nutrient intake, biomarkers or food purchases. The targeted F&V subsidies with nutrition education were able to increase F&V intake by 1–2 serves/day in women. RCT: supermarket price discount of 12.5% significantly increased purchases of total, F&V and healthier food after 6 months.	Quality of evidence is limited because of the risk of selection bias and residual confounding as possible explanations of the success in the 10 non-randomised studies. Limited evidence of success In adults and children. Much of the dietary intake data comes from self-report (imprecision). Follow-up is present in all studies, but in only two success is maintained significantly at the end of the follow-up period: only two studies reported on follow-up post-intervention and after subsidies of six months duration and 12 months and found persistence of effect. Lack of definition of the optimal duration of the intervention.	9
An, 2013 [22]	Evidence of success in 19/20 studies. Level of subsidies varies from 10% to 50% and voucher’s value ranges 7.50$ and 50$ (except for an intervention in which voucher’s value is 0.50$). The comparison between interventions’ results are neither agree with each other, nor exhaustive.	Use of small sample/convenience sample, in specific targeted setting, short time and often no follow-up (in seven it is present, but the results are not maintained). Lack of cost-effectiveness analysis and evaluation of the impact on food industry. Lack of analysis of the impact on the whole diet pattern and not only on targeted foods. Poor evidence of the impact on different population subgroup (ethnicity, SES…).	5
Purnell, 2014 [23]	Subsidies show a short-term positive effect on diet pattern: change is not maintained after a follow-up period.	Heterogeneity of studies (setting, population, methods) makes comparison difficult.	5
Afshin, 2017 [24]	Evidence of success (class I, level of evidence A, AHA evidence grading system). A 10% price decrease increases the consumption of the targeted food of 12%, of the food groups –fruits and vegetables of 14% and of other foods of 16%. The association of taxes and subsidies increases the positive effect.	Most outcomes are evaluated through a self-questionnaire (less strong than objective sales’ measure).	8
Gittelsohn, 2017 [25]	Evidence of success. Intervention’s impact at retail level is an increased sale of targeted healthy foods (from 15% to 1000%; increased purchases in eight), of stocking (from 40% to 63%; increased consumption in 13). Three studies found no effect on healthy foods purchase and consumption.	No strong indication to affirm that one type of intervention is better than another is. All seemed to be effective.	10 *
Alagiyawanna, 2015 [26]	Evidence of success with limits. In CBA: statistically significant evidence of success between subsidies and both F&V and other healthy foods intake in HICs. Effects after 6mo of follow-up. In natural experiment: 20% reduction in soft drinks tax determined 6.8% increase in average consumption.	Lack of RCTs. Use of self-reported information about diet change: bias and errors.	8 *
Hillier-Brown, 2017 [27]	Weak to moderate evidence of effectiveness, in increasing purchase of healthy food (three about incentives).	Low quality evidence with few high-quality designs. Impact seems to be insignificant. Limited generalizability of the results because data come from specific fast food chains within USA, and no info at food outlet level and on consumption. Lack of information about change in consumption, only about purchases. Lack of cost-effectiveness evidence reported.	10 *

* score attributed according by the Authors using the Health Evidence tool when not yet assigned.

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
