# Peer review of "An Umbrella Review and Narrative Synthesis of the Effectiveness of Interventions Aimed at Decreasing Food Prices to Increase Food Quality"

_ijerph, 2019, doi:10.3390/ijerph16132346_

Round 1

Reviewer 1 Report

Dear Authors,

this is a good paper on an important topic, and I have only some minor comments:

Title of the paper: I suggest "A Meta-analysis on the Effectiveness of Fiscal Measures on Dietary Choices"

Abstract: Could be compressed.

1. Introduction: In the last paragraph, refine the aim of the study -  not to provide evidence but seek and address the evidence provided by previous systematic reviews in order to get a fair idea of the state of the art. 

3. Results: Change to Search Results. 

3.1 Description of the selected studies: Display the reviews in similar order in Tables 3-4.

3.2 Interventions: Change to Findings. Discuss the relevance of observed increase in consumption or purchases in the light of the wanted outcome, that is dietary improvement. This is important, because a notable price decrease of F&V causes also a positive income effect, which may increase the consumption of fat food, too (since they are often complements rather than substitutes) so that the diet may not improve so much after all. You already have this theme (also in chapter 4), but it is worthy of highlighting.  

3.3 Quality assessment of the studies: Delete the subtitle and integrate the quality scores of Table 5 in Table 3 or Table 4 (in one column with the Attributed score implicated by *).

5. Conclusions: Concentrate more on what the meta-analysis was able to give and what are its policy implications. Then, you can suggest more appropriate approaches for further studies in the field. And don't use citations here - the WHO one suits better to the Introduction, and the second one is unnecessary.  

Overall: Please check spelling and typos (there are quite many), and please open all acronyms (there are too many). 

Reviewer 2 Report

The article deals with an important topic, and the method is interesting.

I appreciate the depth of analysis, and providing also such insights as "Nevertheless, many studies used a fixed, non-variable, level of subsidy that does not let us identify a dose-response relationship for different populations and time spans."

But the article mentions habits only once - line 169. Overall, there is a static views.
Physicists tend to say that some energy is needed for the system to "find" another equilibrium. In other words, a deep discount or another strong intervention is needed for a reasonably long time in order to change habits, and once this is established, it is possible to somewhat increase prices or stop with an information campaign.
In other words, the article lacks a proper discussion on the minimal necessary length of interventions.

The article could mention also expected unintended consequences, such as if fruit is cheap enough, people may use it to produce wine; and when vegetables and milk are cheap enough, village people may start to use it to feed pigs.

Another unintended consequence could be that lowering price of some healthy food may decrease sales of healthy food with an unchanged price.
In my opinion, it is important to point out also this, so policy makers take it into account and deal with expected unintended consequences - so companies providing healthy food are not affected while other companies providing healthy food are helped.
In other words, e.g. interventions involving lower price of fruits and vegetables should not decrease sales of sesame seeds.

Another limitation of any intervention is the actual quality of available food. In some countries, avocados are sold long before they are ripe. Even if their price drops to 1%, plenty of people will not be interested. Some berries are spoiled 36 hours after they arrive to stores. There can be mold on the bottom of boxes with mushrooms within 4 days but they are kept on shelves for a week.

The article could provide more insights with regards to boundary conditions, so a policy maker reading this article could suggest a law that would not need to be revised after 6 months.
